# Nutritional, Physico-Chemical, Phytochemical, and Rheological Characteristics of Composite Flour Substituted by Baobab Pulp Flour (*Adansonia digitata* L.) for Bread Making

**DOI:** 10.3390/foods12142697

**Published:** 2023-07-13

**Authors:** Sylvestre Dossa, Monica Negrea, Ileana Cocan, Adina Berbecea, Diana Obistioiu, Christine Dragomir, Ersilia Alexa, Adrian Rivis

**Affiliations:** 1Faculty of Food Engineering, University of Life Sciences “King Mihai I” from Timisoara, Calea Aradului No. 119, 300645 Timisoara, Romania; sylvestredossa04@gmail.com (S.D.); ileanacocan@usvt.ro (I.C.); christine.dragomir98@gmail.com (C.D.); ersiliaalexa@usvt.ro (E.A.); adrianrivis@usvt.ro (A.R.); 2Faculty of Agriculture, University of Life Sciences “King Mihai I” from Timisoara, Calea Aradului No. 119, 300645 Timisoara, Romania; adina_berbecea@usvt.ro (A.B.); dianaobistioiu@usvt.ro (D.O.)

**Keywords:** nutritional, organoleptic, antioxidant activity, fatty acids, flavonoids, MIXOLAB, total polyphenols content (TPC)

## Abstract

The aim of this paper is to improve the nutritional quality of bakery products by replacing wheat flour (WF) with different proportions (10%, 20%, and 30%) of baobab flour (BF). The composite flours and bread obtained were evaluated from nutritional, physical-chemical, phytochemical, organoleptic, and rheological points of view. The results obtained show that BF is a rich source of minerals (K: 13,276.47 ± 174 mg/kg; Ca: 1570.67 ± 29.67 mg/kg; Mg: 1066.73 ± 9.97 mg/kg; Fe: 155.14 ± 2.95 mg/kg; Na: 143.19 ± 5.22 mg/kg; and Zn: 14.90 ± 0.01 mg/kg), lipids (1.56 ± 0.02 mg/100 g), and carbohydrates (76.34 ± 0. 06 mg/100 g) as well as for the phytochemical profile. In this regard, the maximum contents for the total polyphenols content (TPC) were recorded in the case of bread with 30% BF (297.63 ± 1.75 mg GAE/100 g), a total flavonoids content (TFC) of 208.06 ± 0.002 mg QE/100 g, and 66.72 ± 0.07% for antioxidant activity (AA). Regarding the physical-chemical, rheological, and organoleptic analysis, the bread sample with 10% BF (BWB1) was the best among the samples with different proportions of BF. It presented a smooth, porous appearance (73.50 ± 0.67% porosity) and an elastic core (85 ± 0.27% elasticity) with a volume of 155.04 ± 0.95 cm^3^/100 g. It had better water absorption (76.7%) than WF (55.8%), a stability of 5.82 min, and a zero-gluten index. The scores obtained by BWB1 for the organoleptic test were as follows: Appearance: 4.81; color: 4.85; texture: 4.78; taste: 4.56; flavor: 4.37; and overall acceptability: 4.7. This study shows that BF improved the nutritional quality of the product, organoleptic properties, α-amylase activity, viscosity, and phytochemical profile, resulting in composite flour suitable for the production of functional bread.

## 1. Introduction

Wheat flour baked goods are a staple food in many countries and therefore have global importance in international nutrition [1]. Wheat flour (*Triticum vulgare*) is an ancient and important flour used for bread making because of the unique baking qualities that it imparts to the dough, namely, extensibility and a viscoelastic structure, which is related to the presence of its constituent gluten protein [2]. It provides essential carbohydrates and some protein but does not provide sufficient minerals, especially calcium, zinc, and iron [3,4]. This has led to the development of functional bakery products with high nutritional value and the right balance of proteins, lipids, minerals, and carbohydrates, helping to achieve adequate intake values to maintain and improve the health of the population [4,5,6]. Bread is one of the main products in the bakery sector, and it is considered an excellent input for micronutrient enrichment and functional ingredients in foods, thus promoting added value [7].

Most parts of the baobab are edible and are processed into other products. The fruit pulp is eaten raw or processed into juices, jams, and sweets; the seeds are used as a thickener for soups and stews; the oil pressed from the seeds is used for consumption and cosmetics. Leaves are used fresh or dried in soups and sauces [8,9].

Several types of research have established an increased demand and consumption of functional bread due to its therapeutic benefits [10]. Therefore, bread can be considered an ideal matrix through which functionalities could be provided to consumers with attractive organoleptic properties.

Baobab (*Adansonia digitata* L.) is a multipurpose tree native to the semi-arid and sub-humid areas of sub-Saharan Africa. It is a long-lived tree (several hundred to thousands of years) and has multiple uses, with more than 300 traditional uses listed [8,9,11,12,13]. The pulp of the baobab fruit adds value to the diet due to its high mineral and vitamin contents. According to FAO/WHO Codex regulations, the baobab fruit pulp is a “high source” not only of vitamins but also of calcium (Ca) and magnesium (Mg), as the fruit provides more than 30% of the Nutrient Reference Value for the labeling of the respective nutrients (the Nutrient Reference Values for the labeling of calcium and magnesium are 800 and 300 mg/100 g, respectively) [14]. The proximate composition and mineral concentration of the pulp and seeds were determined [15]. Considerable fiion, potassium (K), calcium (Ca), magnesium (Mg), sodium (Na), phosphorus (P), ferrum (Fe), and zinc (Zn) contents were observed by Hyacinthe et al. [16].

Baobab contains organic acids and phenolic compounds [8,9,17,18]. Used in the diet of rural communities in West Africa, baobab fruits can help to improve the quality of the diet by diversifying the local diet, which is poor in many remote areas of this part of Africa [19,20,21]. In 2008, the European Union Commission approved baobab fruit pulp as a novel food ingredient [22]. Since then, the export of baobab fruit powder has increased, mainly to Europe, Canada, and the United States [13].

The objective of this article is to study the nutritional, physicochemical, organoleptic, and rheological potential of baobab flour and baobab/wheat composite flour in different proportions.

## 2. Materials and Methods

### 2.1. Materials, Reagents, and Instruments

Baobab flour (BF) was acquired in the North of Benin, and wheat flour (WF) type 650 was acquired at the Profil supermarket, Timisoara (Romania).

The reagents used were DPPH solution and HCl (Sigma-Aldrich; Merck KGaA, Darmstadt, Germany), Ethyl alcohol (SC Chimreactiv SRL, Bucharest, Romania), Folin-Ciocâlteu reagent (Sigma-Aldrich Chemie GmbH, Munich, Germany), and Na_2_CO_3_ (Geyer GmbH, Renningen, Germany). The equipment used in the study was Specord 205 (Analytik Jena AG, Jena, Germany), Chopin Mixolab (Chopin Technologies, Paris, France), the Varian 220 FAA spectrophotometer (Palo Alto, CA, USA), and Thermostat INB500 (Memmert GmbH, Schwabach, Germany).

### 2.2. Preparation of Composite Flours

The substitution levels were determined according to Górecka et al. and Mounjouenpou et al. [23,24]. Three types of composite baobab flour were made: BWF 1 (10% baobab flour and 90% wheat flour); BWF 2 (20% baobab flour and 80% wheat flour), and BWF 3 (30% baobab flour and 70% wheat flour).

### 2.3. Bread Preparation

The bread was prepared according to Hernandez-Aguilar C. et al. [25] and Plustea L. et al. [26], with some modifications. All ingredients (honey, wheat flour type 650, salt, oil, and pakamaya yeast) used to produce the bread, apart from baobab flour, were purchased from the local supermarkets Profile, Timisoara, Romania. Three experimental breads (BWB1, BWB2, and BWB3) were prepared by supplementing a control bread (WB) with different amounts of BF (WB—wheat bread, BWB 1—baobab/wheat bread 10%, BWB 2—baobab/wheat bread 20%, and BWB 3—baobab/wheat bread 30%). The composition of the different breads is shown in Table 1. The technological scheme for obtaining bread with composite baobab flour is presented in Figure 1.

The yeast suspension was obtained by mixing yeast, flour, honey, and salt in 300 g of warm water (30 °C) for 5 min in a mixer with a spiral hook at an 80 rpm speed. At the second speed of 160 rpm, which lasted 5 min, the oil was added gradually. The obtained dough was then placed in a covered plastic bowl for 1 h at a temperature of 20 °C. Then, the dough was kneaded and placed in a pan greased with butter, where it was left to rise for 30 min at a temperature of 35 °C. Afterwards, the dough was baked simultaneously in a preheated electric oven at 230 °C for 24 min. After baking, the bread was left to cool down at room temperature for 24 h and then cut into slices with a knife [25].

The breads obtained from this study are presented in Figure 2.

### 2.4. Determination of Proximate Composition

The approximate composition was determined by the following methods:

Ash content (%): ISO Method No. 2171/2007; Moisture and protein content (%): ICC Standard methods (2003) [27]; and fat content (%): AOAC (2000) [28].

The carbohydrate content (g/100 g) and nutritional value (kcal/100 g) were determined according to [26] using Equations (1) and (2), as shown below:Carbohydrate (g/100 g) = 100 − (fat + protein + water + ash)(1)
Energy value (kcal/100 g) = (fat × 9) + (carbohydrate × 4) + (protein × 4) (2)

### 2.5. Macro and Microelements

The macro- and microelements content was determined by atomic absorption spectroscopy (AAS) using the Varian 220 FAA equipment, according to the method described by [26]. The results were expressed in mg/Kg.

### 2.6. Physical-Chemical Properties

For each type of bread, the following physicochemical characteristics were used to assess quality: volume, porosity, core elasticity, height/diameter ratio, moisture, and acidity. All analytical methods used were carried out according to STAS 91/83 [29]

Briefly, the bread volume was determined with the FORNET device, it is based on measuring the volume of rapeseed displaced by the product analyzed, and it is reported in terms of a percentage. The bread volume is calculated with the following formula:*Volume* (*V*) *=* (V1/m) × 100 (cm^3^/100 g)(3)

The determination of elasticity consists of pressing a piece of core in a cylinder form with parallel sides and a thickness of 60 mm for 1 min and measuring the return to the initial shape after stopping the pressing. The elasticity of the core is calculated with the formula:*Elasticity* (*E*) = (B/A) × 100(4)

(A—height of the core cylinder before pressing; B—height of the core cylinder after pressing and its return to the initial position).

The determination of porosity is the pore volume found in 100 g of the core. The method is based on determining the specific mass of the pore-free core. The porosity is expressed in terms of volume percentages and is calculated with the formula:*Porosity =* [*V* − (*m*/*ρ*)/*V*] × 100, [% vol](5)

(*V*—the volume of the core cylinder, in cm^3^; *m*—the mass of the core cylinder, in g; *ρ*—the density of the compact core, in g/cm^3^).

The product height and diameter are measured, and their ratio is calculated with the formula:*Height/diameter ratio = H/D*(6)

(*H*—maximum height of the product, *D*—arithmetic mean of two perpendicular diameters, in centimeters).

The values obtained are the arithmetic means of the two parallel analyses.

### 2.7. Phytochemical Profile

#### 2.7.1. Preparation of Alcoholic Extracts

From each sample of composite flour and bread, 1 g was weighed into lidded containers, to which 10 mL of 70% (*v*/*v*) ethanol (Chimreactiv, Bucharest, Romania) was added. The containers were hermetically sealed and shaken with a magnetic stirrer (IDL, Freising, Germany) for 30 min, after which they were filtered through Whatman N°1 filter paper. The extracts thus obtained were then used to determine the total polyphenol content, total flavonoid content, and total antioxidant activity by DPPH.

#### 2.7.2. Evaluation of the Total Phenolic Content (TPC)

The determination of the total phenolic content of BF, WF, composite flours, and bread with different percentages of baobab flour was done according to the method of Folin-Ciocâlteu [30,31] with some modifications. The results were reported as mg gallic acid equivalent (GAE) per 100 g sample. All determinations were performed in triplicate.

#### 2.7.3. Determination of Total Flavonoid Content (TFC)

The total flavonoid content (TFC) was analyzed according to the modified method described by Cocan et al. (2022) [32]. In total, 1 g of each sample and 10 mL of 60% ethanol were mixed. The containers were closed and placed in the Holt plate stirrer for 30 min. They were then filtered through filter paper. A total of 1.5 mL of the previously prepared extract was added to 4.5 mL H_2_O and 1 mL NaNO_2_ and incubated for 6 min. After the incubation period, 1 mL of 10% Al(NO_3_)_3_ was added and incubated again for 6 min. After incubation, 10 mL of 4% NaOH was added and made up to 20 mL with 70% alcohol.

The samples were left to stand for 15 min. After 15 min, the absorbance was read at 510 nm using a UV-VIS spectrometer (Analytical Jena Specord 205, Jena, Germany). Quercitin solutions (QE) were used as the standard. The results were expressed as mg EQ/100 g, and all determinations were performed in triplicate [32].

#### 2.7.4. Antioxidant Activity

The antioxidant activity of baobab flour, wheat flour, and composite flours, as well as bread made from these flours, was determined by 2,2-diphenyl1-picrylhydrazyl (DPPH) using the method described by Ciulca et al. [33], with some modifications. We added 1 mL (*v*/*v*) of the extract (1/10) and 2.5 mL of a 0.03% mM DPPH solution. The resulting mixture was then incubated for 30 min at room temperature in a dark place. Following this, the absorbance was read using a spectrophotometer (Specord 205; Analytik Jena AG, Jena, Germany) at 518 nm. Ethyl alcohol was used as the positive control. The antioxidant activity was determined by the following equation:RSA (%) = (Acontrol − Asamples/Acontrol) × 100
where Acontrol denotes the control absorbance values, and Asamples denotes the samples’ absorbance values

All measurements were performed in triplicate.

### 2.8. Rheological Analysis

Rheological analysis was conducted to evaluate the effect of replacing wheat flour with baobab flour (10%, 20%, and 30%) on the rheological properties using Chopin Mixolab equipment (Chopin Technologies, Paris, France) and the “Chopin+” protocol [34].

Quantities ranging from 42 to 50 g of the sample (depending on the sample’s moisture content) were placed in the Mixolab bowl and mixed. After tempering the solids, water was added to achieve optimal consistency. Particular attention was paid to determining water absorption to ensure the complete hydration of all the components.

The parameters taken into consideration from the Mixolab profile were: water absorption, dough development time, stability (mixing resistance of dough), maximum torque during mixing—C1, weakening of the protein—C2, which manifests as a result of mechanical stress as the temperature rises, the rate of starch gelatinization—C3, minimum torque—C4, and torque—C5 after cooling at 50 °C. In addition, Mixolab determined the following parameters: cooking stability (C4/C3), protein weakening under a heating effect (alpha slope), starch gelatinization speed (beta slope), enzyme degradation speed (gamma slope), and starch retrogradation at the cooling stage (C5–C4), which represents the shelf-life of the final products [34].

### 2.9. Sensory Analysis

A panel of 27 evaluators, aged between 20 and 52 years, who were non-smokers and had no known cases of food allergies, evaluated the four bread samples obtained (WB: control wheat bread; BWB1: 10% baobab flour and 90% wheat flour; BWB2: 20% baobab flour and 80% wheat flour; BWB3: 30% baobab flour and 70% wheat flour).

All 27 panelists were trained according to ISO 6658:2017 [35]. This standard was also used to conduct this evaluation based on a five-point hedonic scale:

1 = extremely disliked;

2 = slightly disliked;

3 = neither liked nor disliked;

4 = slightly liked;

5 = extremely liked.

The bread was cut into 1 cm thick slices, coded in two-digit characters, and served in random order under normal lighting conditions and at room temperature.

The acceptability score and level ranges were as follows: 1.00–1.49 = Not Acceptable (NA); 1.5–2.49 = Slightly Acceptable (SA); 2.50–3.49 = Moderately Acceptable (MA); 3.5–4.49 = Acceptable (A); and 4.5–5.00 = Very Acceptable (HA) [26].

### 2.10. Statistical Analysis

All determinations were performed in triplicate. The results are presented as the mean values ± standard deviation (SD). The differences between the means were analyzed by a *t*-test (two samples assuming equal variances) with Microsoft Excel 365. Significant differences were considered when the *p*-values were less than 0.05.

## 3. Results and Discussion

### 3.1. Determination of Proximate Composition

The results presented in Table 2 illustrate the characteristics of the composite baobab flours.

The analysis of the results (Table 2) shows that BF is richer than WF in terms of the mineral substances, lipid, carbohydrate, and water concentrations, whereas WF is richer in protein and provides more energy than BF. The moisture content increased from 10.80 ± 0.04% (WF) to 11.80 ± 0.03% (BWF3) for the flours and from 35.20 ± 0.02% (WB) to 34.73 ± 0.04% (BWB3) for the bread. There was an observed increase in the moisture content with the addition of BF both in the composite flours and in the bread with different percentages of BF, in accordance with the findings of Barakat H. et al. [36]. Our results also show that BF is more than 12 times richer in minerals (4 ± 0.02%) than WF (0.33 ± 0.03%). The value obtained here for mineral substances for BF is slightly lower than that obtained by [36,37,38,39,40] (5.2 to 6.52%). This difference between the results is explained by the fact that the nutritional composition of baobab varies from one region to another [40]. Compared to composite flours, we have 0.66 ± 0.02%, 1.1 ± 0.18%, and 1.21 ± 0.04% of mineral substances in BWF1, BWF2, and BWF3 respectively. This means that as the amount of BF increases in the composition of the composite flours, the number of minerals increases. The mineral content in the breads obtained also showed the same behavior, increasing with the amount of baobab in the bread (0.96 ± 0.02%, 1.69 ± 0.01%; 1.95 ± 0.03%, and 2.17 ± 0.05%, respectively, for WB, BWF1, BWF2, and BWF3). These results show that the partial substitution (up to 30%) of wheat with baobab would increase the mineral content both in the composite flours and in the bread produced with these flours. In the results of [36], the increase in BF is in agreement with the mineral abundance, thus confirming our results.

The protein content of WF is significantly higher (*p* < 0.05) than that of BF (11.26 ± 0.02% versus 4.31 ± 0.05%). It can be seen that as the substitution rate of BF increases in the composition of both composite flours and the breads, the protein content in the composite flours as well as in the breads decreases, in agreement with other authors [36]. The protein content of the baobab pulp obtained in other studies ranged from 2.04 to 17% [35,36,37,38,41,42,43]. Concerning lipids, it should be noted that BF has a higher content than WF (1.56 ± 0.02% versus 1.33 ± 0.03%). Slightly lower results than ours (from 0.4 to 0.94%) were obtained for the lipid content of BF [36,37,38]. A slight increase in the lipid content in the composite flours and in the bread was observed with the progressive addition of baobab flour. The results are 1.40 ± 0.05, 1.42 ± 0.03, and 1.45 ± 0.01 for BWF1, BWF2, and BWF3 for composite flours and 5.19 ± 0.03, 5.20 ± 0.05, 5.26 ± 0.02, and 5.30 ± 0.02 for BW, BWB1, BWB2, and BWB3 for breads, respectively. From the analysis of these results, we can say that the enrichment of wheat bread with BF would slightly improve the lipid intake in the bread, which is not the case for the protein content. In the studies of [36], the lipid and protein content did not vary significantly from one cake sample to another. In contrast, in the studies of [24], there was an increase from one sample to another. As in the case of lipids, our results show that BF is slightly richer in carbohydrates than WF (76.34 ± 0.06% versus 76.28 ± 0.04%). According to our results, baobab would provide a slight source of carbohydrates to the composite flours as well as to the bread made from these flours. Contrary to this, we note that the energy value provided by BF (336.62 ± 0.16 kcal/100 g) and the composite flours (from 358.80 ± 0.22 to 355.21 ± 0.23 kcal/100 g) is lower than that of WF (362.13 ± 0.16 kcal/100 g), and on the other hand, that provided by bread with different percentages of BF (from 280.84 ± 9.5 to 278.93 ± 0.41 kcal/100 g) is below that provided by WB (281.31 ± 0.2 kcal/100 g). Our results thus reveal that BF would provide a lower energy source than WF. Our results are in agreement with those of [24,36]. In the article of [35], from 351.75 ± 17.59 kcal/100 g in the control sample, the energy value increased to 345.75 ± 12.10 kcal/100 g in the cake sample with 15% BF. In studies conducted by Mounjouenpou P. et al., [24] the energy value of biscuit samples with different percentages of baobab ranged from 490.24 kcal/100 g (20% BF) to 505.24 kcal/100 g (control biscuits).

### 3.2. Macro and Microelements

In Table 3, the contents of the following macroelements are presented: potassium (K), calcium (Ca), magnesium (Mg), and sodium (Na), along with those of the following microelements: the ferrum (Fe), zinc (Zn), cuprum (Cu), manganese (Mn), and nickel (Ni) of the wheat flour, the composite flours, and the bread obtained.

The results reveal that potassium (K) is the most abundant nutrient in baobab flour (13,276.47 ± 174 mg/kg). This result is in agreement with those of [36,37,38,40,43], where potassium (K) was the most abundant macronutrient in BF. Moreover, the value we found is in the range of those obtained by [36,37,38,40,43], i.e., between 9.875 and 23.90 mg/g, against 13.276 mg/g in our case. According to our results, potassium (K) is also the most abundant nutrient in composite flours as well as in bread with different proportions of BF. Furthermore, we can see that there is a significant difference in the potassium (K) content between WF and the other flours with different BF proportions (BWF1, BWF2, and BWF3), on the one hand, and between the control bread (WB) and the other bread with different BF proportions (BWB1, BWB2, and BWB3) on the other. It was also found that the higher the amount of BF in the composition of both the flour and the bread, the more abundant K was. All these findings were also reported by [24,36]. Potassium (K), as a mineral, together with sodium (Na), helps to maintain the fluid balance within the cells as well as the pH. It therefore contributes to the stability of blood pressure and is necessary for muscle contractions (e.g., those of the heart muscle) and for the transmission of nerve impulses [44]. The World Health Organization (WHO), through its guidelines on potassium intake in adults and children, recommends an increase in dietary potassium intake to lower blood pressure and reduce the risk of cardiovascular disease, stroke, and coronary heart disease in adults. It also recommends increasing dietary potassium intake to combat high blood pressure in children. It therefore suggests that one’s potassium (K) intake should be at least 3510 mg/day in adults. However, in children, this value should be revised downwards according to the energy needs of the age group [45]. From all this information, we can say that the consumption of 100 g of our bread with 30% baobab (BWB3) alone would meet almost 6% of the daily potassium requirement.

As far as calcium (Ca) is concerned, there is a significant difference in its values between WF and the other types of flours. There also a significant difference between WB and the other types of bread. Calcium is more abundant in BF (1570.67 ± 29.67 mg/kg) than in WF (166.22 ± 0.99 mg/kg). The calcium (Ca) level in BF in our study is slightly lower (237.03 to 370 mg/100 g) than in other studies [36,37,38,39,40,41,42,43]. Further, 246.12 ± 2.62 mg/kg, 307.95 ± 2.05 mg/kg, and 329.15 ± 6.40 mg/kg are the calcium (Ca) compositions of BWF1, BWF2, and BWF3 (composite flours) respectively; 327.49 ± 1.05 mg/kg, 343.13 ± 8.61 mg/kg, 366.05 ± 3.55 mg/kg, and 395.05 ± 2.85 mg/kg are the values for BWB1, BWB2, and BWB3 (breads obtained), respectively. It can be seen that the more BF is abundant, the more calcium (Ca) is abundant in both the flour and the bread. With the same remark being made for K, we can deduce that the partial substitution of WF with BF would improve the calcium (Ca) and potassium (K) composition in the composite flours as well as in the bread derived from them. These results correspond to those of [24,36].

According to the European Food Safety Authority (EFSA) Scientific Panel on Dietetic Products, Nutrition, and Allergies, calcium is an essential macro-element for healthy bones and teeth. It is important for neuronal transmission, coagulation, muscle contraction, cell signalling, and many other functions. It is the fifth-most abundant element in our body. Almost all (99%) of the body’s total calcium is found in the bones and teeth, mainly in the form of calcium hydroxyapatite. Although calcium is important for our body, its excess leads to serious consequences such as nephrolithiasis and impaired renal function, resulting in a loss of the kidney’s ability to concentrate (i.e., a decrease in salt and water reabsorption), as well as in volume and sodium depletion [46]. Finally, the EFSA Scientific Panel on Dietetic Products, Nutrition, and Allergies recommends a daily calcium intake not exceeding 2500 mg/day for adults, including pregnant and lactating women. The consumption of 100 g of BWB3 alone would provide almost 40 mg of Ca, i.e., about 2% of the daily requirement for adults, including pregnant and breastfeeding women.

Magnesium (Mg), like calcium, is an essential macro-element for the functioning of the human body. It plays an important function in more than 300 metabolic reactions in the body. In particular, it contributes to nerve transmission and muscle relaxation after contraction, which is vital for heart function. It is therefore essential for maintaining a regular heartbeat, for fat metabolism, and for regulating blood sugar and blood pressure. Magnesium (Mg) also contributes to the relaxation of menstrual complaints in women through its relaxing effect on the vessels, dilating muscles, and nerve conduction. About half of the body’s magnesium is found in the bones and teeth, while the rest is located in the muscles, liver, and other soft tissues. It is eliminated by the kidneys [47,48]. The Scientific Panel on Dietetic Products, Nutrition, and Allergies (NDA Panel) has established an adequate intake (AI) for magnesium (Mg) of 350 mg/day for men and 300 mg/day for women. For children, the AI varies from 170 to 300 mg/day, depending on age. Our results show that WF (273.15 ± 1.28 mg/kg) is less abundant in magnesium (Mg) than BF (1066.73 ± 9.97 mg/kg). Thus, an increase in Mg composition is observed when BF is added to both flour and bread. These findings lead us to deduce that the partial substitution of WF with BF results in composite flours and bread with higher magnesium (Mg) contents. We agree with [24,35] because, in their studies, the higher the percentage of BF, the more abundant the Mg content in the finished products. It should be noted that the consumption of 100 g of BWB3 provides 46.478 mg, i.e., 13.28% of the adequate intake recommended by the EFSA NDA group for men, 15.49% of that for women, and between 15.49 and 27.34% of that for children. Contrary to the case of magnesium (Mg), a decrease in the composition of Manganese (Mn) was observed when adding BF to both flour and bread. This is justified by the fact that WF is more abundant in manganese (Mn) (8.69 ± 0.02 mg/mg) than BF (4.84 ± 0.05 mg/kg). It can be deduced that the partial substitution of WF with BF results in composite flours and bread with a higher magnesium (Mg) abundance but lower manganese (Mn) abundance.

Ferrum (Fe) deficiency (anemiain the body, is a public health problem in both industrialized and non-industrialized countries, [49,50]. In our study, the ferrum (Fe) composition of baobab flour (155.14 ± 2.95 mg/kg) is slightly higher than its sodium (Na) composition (143.19 ± 5.22 mg/kg). Each of these two matrices (Fe and Na) is more abundant in BF than in WF. Further, 12.13 ± 0.03 mg/kg and 60.38 ± 0.76 mg/kg are, respectively, the ferrum (Fe) and sodium (Na) compositions of WF. Moreover, for each of these two matrices, there is a significant difference between WF and the composite flours and between WB and the other bread. Also, as for the case of most of the other micro- and macronutrients, BF would have an effect on the abundance of ferrum (Fe) and sodium (Na) in the compound flours and the bread obtained. These results are similar to those of Barakat [36]. Except for manganese (Mn), zinc (Zn), copper (Cu), and Nickel (Ni) are, respectively, the least abundant elements in both BF and WF. For BF, we obtained 14.90 ± 0.01 mg/kg, 8.04 ± 0.05 mg/kg, and 0.598 ± 0.002 mg/kg for Zn, Cu, and Ni, respectively. These values are close to those of [36,37,38,39,40,41,43]. For WF, we have 11.40 ± 0.01 mg/kg and 2.4 ± 0.002 mg/kg Zn and Cu, respectively. It is important to note that the partial substitution of WF with BF would improve the abundance of zinc, copper, and nickel [24,36].

### 3.3. Physical-Chemical Proprieties of Baobab Bread

The different pieces of bread obtained were subjected to various physical-chemical analyses. These were the volume, porosity, elasticity, height/diameter (H/D) ratio, and acidity. The results obtained are presented in Table 4.

An analysis of the data obtained from this table indicates that, apart from acidity, WB recorded the highest values in terms of the volume (190.97 ± 1.05 cm^3^/100 g), H/D ratio (0.62 ± 0.01), porosity (78.84 ± 0.56%), and elasticity (90 ± 0.15%) compared to the other samples. We also note a decrease in the value of each of these parameters if the quantity of BF increases in the composition. Baobab would thus have an influence on the physical-chemical quality of the bread obtained.

As the volume of WB was higher compared to that of the other bread samples with different proportions of BF, it should be noted that the latter was not excessively flattened (Table 4). They varied between 155.04 ± 0.95 and 121.88 ± 1.16 cm^3^/100 g. The BF would therefore have an influence on the volume of the final product obtained. Regarding the H/D ratio, samples with different proportions of baobab flour recorded 0.58 ± 0.02, 0.57 ± 0.01, and 0.55 ± 0.03 for BWB1, BWB2, and BWB3, respectively. According to STAS 91/83 [29], for values of the H/D ratio between 0.4 and 0.6, the bread is considered to be suitable in volume and shape, for those above 0.6, the bread is domed, and a value of the H/D ratio below 0.40 is an unsuitable and flattened product. Thus, we can conclude that all breads have an appropriate volume and shape. Apart from having an appropriate form and volume, these breads were also porous, with a porosity ranging from 73.50 ± 0.67 to 66.22 ± 0.45%. They were also elastic (85 ± 0.27 to 72 ± 0.19%).

Concerning the acidity, it can be seen that it increases with the addition of baobab flour to the composition. It goes from 2.4 ± 0.05° acidity/100 g for WB to 5.4 ± 0.08° acidity/100 g for BWB1 and from 10.8 ± 0.05° acidity/100 g for BWB2 to 15 ± 0.06° acidity/100 g for BWB3. It is therefore concluded that BF has a higher acidity than WF. All these parameters deserve particular attention during the production of bread with BF in order to have a finished product with better physical-chemical characteristics. Following the research, it should be noted that bread with 10% baobab flour (BWB1) has the best physico-chemical characteristics (smooth, porous appearance and elastic core) compared to the samples BWB2 and BWB3.

### 3.4. Rheological Properties of the Different Flours

In order to know the rheological profile, an analysis was carried out using Mixolab. The Mixolab is a device that measures the rheological behavior of dough that is subjected simultaneously to kneading and a temperature. It measures, in real time, the torque (Nm) produced by the dough between the blades. The test is based on the preparation of a constant hydrated dough mass in order to obtain a target consistency in the first test phase [34].

Figure 3 shows the rheological profiles of WF and the different flours with different proportions of baobab flour (10%, 20%, 30%, and 100%), as well as the photo of the different breads obtained.

Table 5 shows the primary parameters of BF, WF, BWF1, BWF2, and BWF3.

The water absorption (WA) is the amount of water that the flour absorbs to achieve a given consistency during the constant temperature phase. It also corresponds to the hydration needed to obtain a maximum dough consistency of 1.1 Nm [34]. According to our results, BF has a better water absorption capacity than WF. It can be seen that WA increases from 55.8% for WF to 62.1% when 10% BF is added to the composite mix and to 63.3% for BWF3, i.e., when 30% BF is added. Thus, the WA increases with the addition of BF. The same observation was reported by Barakat [36] in his work. Indeed, from 59.1 ± 0.9% in his control sample, the WA increased to 64.6 ± 0.5% for the sample with 15% BF. This leads to the conclusion that the substitution of WF with BF gives compound flours better water absorption. This parameter should be given special attention for the production of good-quality finished products.

Concerning dough stability, which refers to the resistance of the dough to kneading, BF is clearly lower than that of WF (0.30 min against 9.52 min). Also, it decreases with the addition of BF in the composite flours. From 5.82 min for BWF 1 through 3.97 min for BWF 2, it finally drops to 3.82 min for BWF 3. This is the exact same observation that was made by [36], who reported that dough obtained from composite flours was less stable than the control sample. Knowing that the longer the dough stability time is, the “stronger” the dough is, it can be deduced that BF has an impact on the stability of the dough obtained from composite flours by making it less strong and less resistant to kneading [34].

The torque C1 is the difference between the maximum torque at 30 °C and the torque at the end of the holding time at 30 °C [26]. In our study, the value of C1 is 1.189 Nm for WF and 1.255 Nm for BF. For the composite flours, it is at the maximum in BWF3 (1.372 Nm) and at the minimum (1.165 Nm) in BWF1. It also increases as the proportion of BF increases in the composite flours. The C2 torque corresponds to the measurement of protein weakening as a function of the mechanical work and temperature that takes place in the second step of the MIXOLAB profile [34]. Compared to WF (C2 = 0.538), baobab composite flours are characterized by a lower C2 torque (0.287 to 0.342). Similar results were obtained by Barakat [36].

The C3 and C4 pairs correspond, respectively, to the measurement of the gelatinization of the starch and the measurement of the stability of the gel formed when hot [36]. From our results, we can see a decrease in C3 and C4 values with the addition of BF in the composite flours. Thus, if the dough obtained from WF has a C3 value of 2027 Nm and a C4 value of 1860 Nm, as the proportion of BF in the composite flours increases, these values decrease. The lowest values are then recorded in the case of BWF3 (C3 = 1.706 Nm; C4 = 1.524 Nm), which highlights the decrease in dough viscosity and the contribution of alpha-amylase by the addition of BF in WF. This finding is identical to that of [36]. The C5 torque measures the retrogradation of the starch during the cooling period. In the present study, C5 values decrease with the addition of WF. Thus, it decreases from 3.289 Nm for WF to 3.054 Nm for BWF1, to 2.565 Nm for BWF2, and to 2.306 Nm for BWF3. The decrease in this parameter with the addition of BF can be explained by its low value at BF (0.487 Nm).

Table 6 presents the index scores defined by the Mixolab Profiler for WF, BF, and the different composite flours. It shows the values for the Water Absorption Index (WAI), Mixing Index (MI), Gluten + Index (GI), maximum viscosity during heating (expressed as the Viscosity Index (VI)), starch stability or Amylolysis Index (AI), and starch retrogradation or Retrogradation Index (RI) [34].

The Mixolab Profiler is a functionality of the Mixolab system that uses the standard ICC protocol N°173 for a complete characterization of flours (protein network, starch, and enzyme activity) and produces a simplified graphical interpretation of the results (see Figure 3) [30].

WAI is a function of flour composition (protein, starch, fiber, etc.) and has an impact on the dough yield. In the case of our study, the WAI was found to be highest in composite flours (8) and lowest in WF (2). According to [34,51], the higher the WAI, the higher the water absorption capacity of the dough. This confirms the result we obtained regarding the water absorption of BF and the increase in the latter in the composite flours with the addition of BF observed in the previous table. Similar results were obtained by [36].

MI gives information on the stability, development time, and weakening of the dough during mixing at 30 °C [34]. GI gives information on the behavior of the gluten during the heating of the dough. MI and GI are maximal in WF but minimal and constant in composite flours. A high value of MI corresponds to a high stability of the dough during kneading [34], so it can be deduced that the dough obtained from WF is more stable than that obtained from the composite flours (BWF1, BWF2, and BWF3). This confirms the conclusion we made above: BF has an impact on the stability of the doughs obtained from the composite flours by making them less strong, less resistant to kneading, and, thus, less stable. Concerning GI, a high value corresponds to a high resistance of the gluten to heating [34]. It is concluded that WF gives dough with a better gluten network than dough obtained from flours with different proportions of BF.

VI is a function of amylase activity and starch quality. It represents the increase in viscosity during heating [34,51]. In our study, the VI value is seven for WF and decreases with the addition of BF in the composition (BWF1: six, BWF2: four, and BWF3: two). A high VI value corresponds to a high viscosity of the paste during heating [31]. It is deduced that WF produces dough with better viscosity than the other flours. AI and RI represent, respectively, the ability of the starch to resist amylolysis and the characteristics of the starch and its hydrolysis during the test [34,51]. AI and RI are equal to eight each for WF and decrease in composite flours. The high values of AI and RI correspond to a low shelf life of the final product [34]. Thus, BF extends the shelf life of the composite flours. The doughs from the composite flours will therefore have a better shelf life than those from WF.

### 3.5. Phytochemical Profile of Composite Flours and Bread

Flour and bread samples were subjected to phytochemical analyses, namely, the total polyphenol content (mg/100 g), total flavonoid content (mg/100 g), and antioxidant activity (%) (Table 7).

The analysis of the data obtained shows that the higher the amount of BF in the different flours or bread, the higher the total polyphenol (TP) content. For both flour and bread samples, there is a significant difference between the samples with different proportions of BF and the control samples. It can also be seen that the highest TP content is found in the flour or bread samples with 30% baobab (BWF3: 296.1 ± 3.29 mg GAE/100 g and BWB3: 297.63 ± 1.75 mg GAE/100 g). This is also the finding of [36]. Indeed, in his study, it was found that the more baobabs in the composition, the higher the polyphenol content. It is also interesting to note that the total polyphenol content of the bread samples is higher than that of the respective flours. This could be explained by the presence of total polyphenols in the honey used as an ingredient [52]. BF is significantly richer in total polyphenol (629.7 ± 0.35 mg GAE/100 g) than WF (176.7 ± 0.69 mg GAE/100 g). A result below ours was found by Balarabe et al. (2019) [53], who reported a content of 502.02 ± 8.70 mg GAE·100 g^−1^. On the other hand, Cisse Ibrahima (2012) [39] reported a higher value (1084 mg GAE·100 g^−1^). This difference between the results can be explained by the variation in the constitution of the baobab depending on its geographical area [40,54].

Concerning the total flavonoid content, it is more than 200 times more abundant in BF than in WF. There is thus a significant difference between the flavonoid contents of BF and WF. We have, respectively, 1.72 ± 0.08 mg/100 g and 213.13 ± 0.08 mg/100 g for BF and WF. In addition, there is a significant increase in the flavonoid value as the proportion of BF increases in the different flours as well as in the bread. Thus, from 70.53 ± 0.08 mg/100 g for BWF1, it increased to 138.97 ± 0.81 mg/100 g for BWF3, and from 44.05 ± 0.81 mg/100 g for BWB1, it reached 208.06 ± 0.002 mg/100 g for BWB 3. The partial substitution of WF with BF is thus at the origin of the abundance of flavonoids in the composite flours and bread obtained compared to the standard samples, thanks to the richness in flavonoids of the baobab [55].

The same observations made for the total polyphenol and total flavonoid contents were also made for antioxidant activity (AA). Baobab flour has a high AA (86.62 ± 0.04%) compared to WF (31.77 ± 0.43%). Thus, in the composite flours, the AA is increasingly higher with the proportion of BF in them. We have, respectively, 82.08 ± 0.04%, 82.86 ± 0.56%, and 84.86 ± 0.01% for BWF1, BWF2, and BWF3. The same pattern was repeated for the bread. We have 31.00 ± 0.09% for WB and, respectively, 58.86 ± 0.01%, 61.66 ± 0.02%, and 66.72 ± 0.07% for BWB1, BWB2, and BWB3. Similar observations were made by Barakat, H. (2021) [36], who pointed out increasing AA and polyphenol contents with the incorporation of a higher baobab proportion in bakery products.

Following all the data regarding the bioactive compounds, it can be highlighted that the substitution of BF by WF significantly increased the phytochemical properties of composite flours and bread. Baobab is therefore an alternative for the formulation of bread products with high antioxidant activity and high polyphenol and flavonoid contents without the use of chemical additives.

### 3.6. Sensory Analysis

To determine the acceptability of the bread, a sensory evaluation was obtained regarding the different pieces of bread (Figure 4). The sensory evaluation was conducted with a panel of 27 evaluators using a five-point hedonic scale. The mean scores for the sensory attributes (appearance, color, taste, texture, flavor, and overall acceptability) of the baobab bread studied (WB: control wheat bread; BWB1: 10% baobab flour and 90% wheat flour; BWB2: 20% baobab flour and 80% wheat flour; BWB3: 30% baobab flour and 70% wheat flour) are presented in the figure below.

The analysis of this figure shows that the control sample (WB) obtained the best scores independently of the evaluation criteria (color: 4.85; taste: 4.56; appearance: 4.81; texture: 4.78; flavor: 4.37; and overall acceptability: 4.70). We agree with this remark [24,35]. Among the bread samples with different proportions of baobab flour (BWB1, BWB2, and BWB3), BWB1 was the most appreciated by the evaluators, followed by BWB2 and BWB3, respectively. Not only was BWB1 the most appreciated among the baobab composite flour samples, but it also obtained values close to those of the control sample (WB). BWB1 was very acceptable (scores between 4.5 and 5) for its appearance (4.52) and color (4.66). It had average scores (4.37, 3.67, and 4.11, respectively, for texture, flavor, and overall acceptability) in the range of 3.5 to 4.49, which indicates that the product is acceptable in relation to the criteria studied. With regard to taste, BWB1 is classified in the medium acceptability category with a score of 3.30. Sample BWB2 was in the acceptance range of 3.5–4.49 (Acceptable) for appearance, texture, and color. For overall acceptability and flavor, BWB2 was in the 3.5–4.49 acceptance range, so it has an average acceptance for both criteria (Figure 4). For this sample, only taste is in the acceptability category 1.5–2.49 (slightly acceptable). Sample BWB3 with 30% baobab flour is the least appreciated by the panel of evaluators. The evaluation criteria with the highest scores were: color (3.85) and appearance (3.52). Texture (3.41) and overall acceptability (2.70) were in the range of 2.5–3.49 and therefore in the medium acceptability category. For this sample, the taste and flavor with the lowest values (1.88 and 2.33, respectively) and in the acceptability range 1.5–2.49 are therefore slightly acceptable. From an overall point of view, it can be noted that the substitution of WF with BF progressively reduces the acceptability of the final product by the consumer. Nevertheless, a substitution of no more than 10% results in a presentable bread that is well appreciated by the consumer because of the flavor and color provided by the baobab.

In Barakat’s [36] studies on the nutritional and rheological characteristics of composite flour substituted with baobab (*Adansonia digitata* L.) pulp flour for cake making and the organoleptic properties of their prepared cakes, he came to the conclusion that the addition of 5–10% BF has no drastic effect on the organoleptic characteristics of the cake, so the addition of 15% was still acceptable. However, at higher substitution levels, the organoleptic characteristics are affected. In contrast, Mounjouenpou et al. [24], in their studies on the effect of fortification with baobab (*Adansonia digitata* L.) pulp flour on the sensory acceptability and nutritional composition of rice crackers, reported that the substitution with 20% BF improved the sensory and nutritional qualities of rice crackers.

Following the results of the sensory analysis, it can be seen that the samples with a higher baobab content scored lower compared to the samples with 10% baobab flour, probably due to the fact that baobab flour adds a tangy and citrusy flavor to the products, which was not to the consumers’ liking.

In the framework of our study, and based on our results, we can say that the incorporation of BF in bread making should not exceed 10% in order to maintain the characteristics of the bread obtained.

## 4. Conclusions

This study highlighted the nutritional, phytochemical, physicochemical, rheological, and organoleptic properties of baobab flour in bread making. It revealed that baobab can be a source of fortification for bakery products, especially bread. The partial substitution of wheat flour for baobab flour has resulted in composite flours and bread with better nutritional compositions, particularly in minerals, than standard bread made with 100% wheat flour. It is also an alternative to reducing gluten in bread, as baobab does not contain any. This study also reveals an improvement in phytochemical parameters due to the substitution of wheat flour with baobab. Nevertheless, with a high substitution rate, although the nutritional and phytochemical values increase, we obtain bread that is less appreciated from the organoleptic point of view. This also affects the physical-chemical and rheological characteristics. Therefore, with a substitution rate not exceeding 10%, the bread obtained is not only rich in nutrients and presentable from the physical-chemical point of view but it is also well appreciated by the consumer and has no negative effects on the rheological and technological characteristics.

## Figures and Tables

**Figure 1 foods-12-02697-f001:**
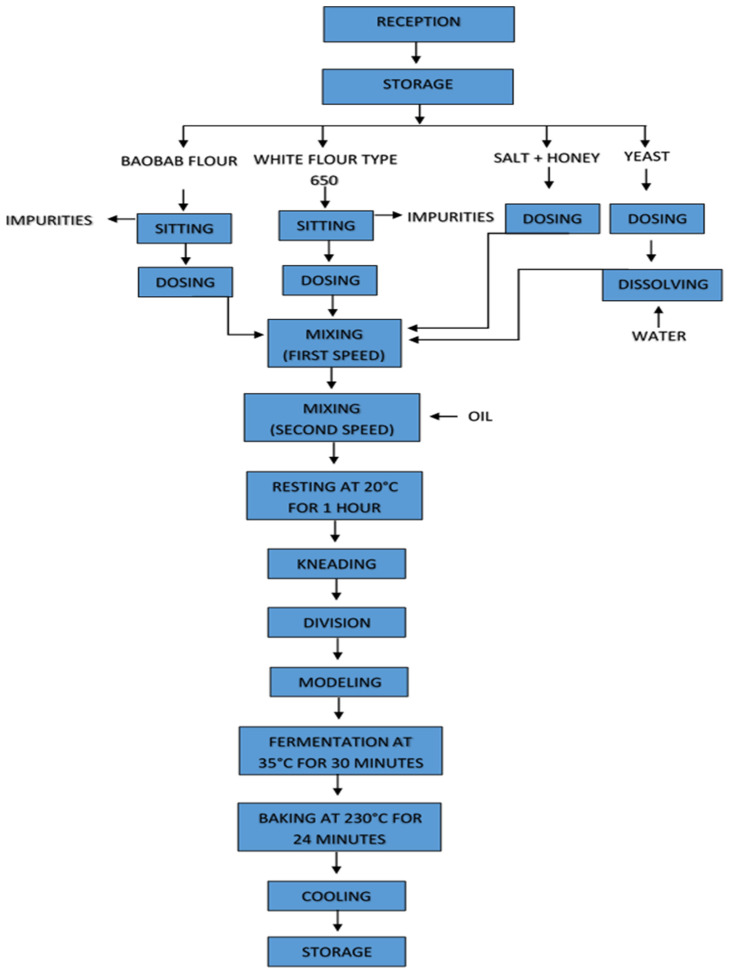
The technological scheme for obtaining bread.

**Figure 2 foods-12-02697-f002:**
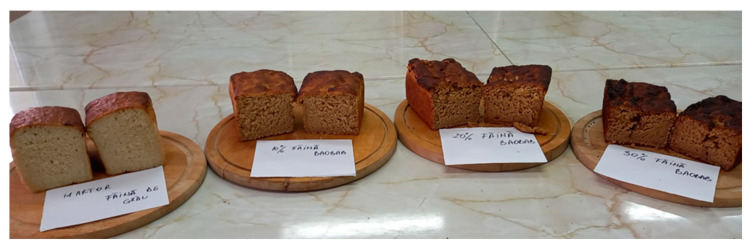
Final products. WB—wheat bread, BWB 1—baobab/wheat bread 10%, BWB 2—baobab/wheat bread 20%, and BWB 3—baobab/wheat bread 30%.

**Figure 3 foods-12-02697-f003:**
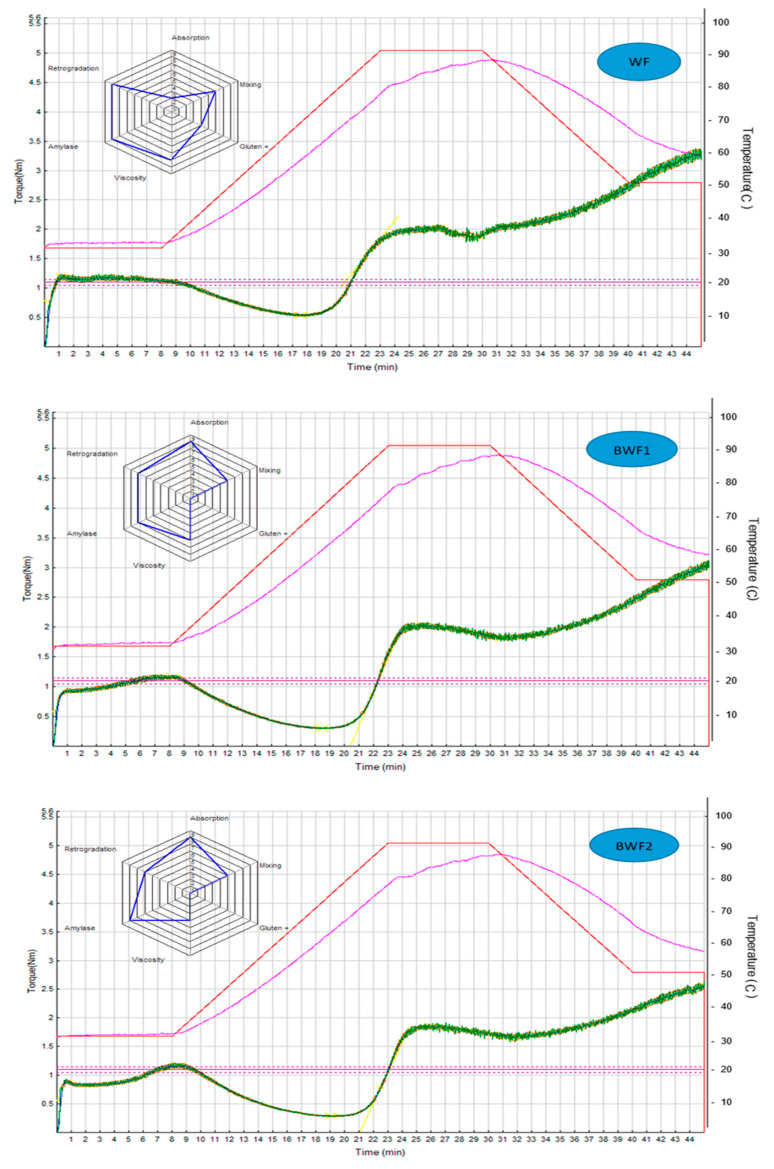
Mixolab rheological profiles of the analyzed flour samples (WF, BWF1, BWF2, and BWF3).

**Figure 4 foods-12-02697-f004:**
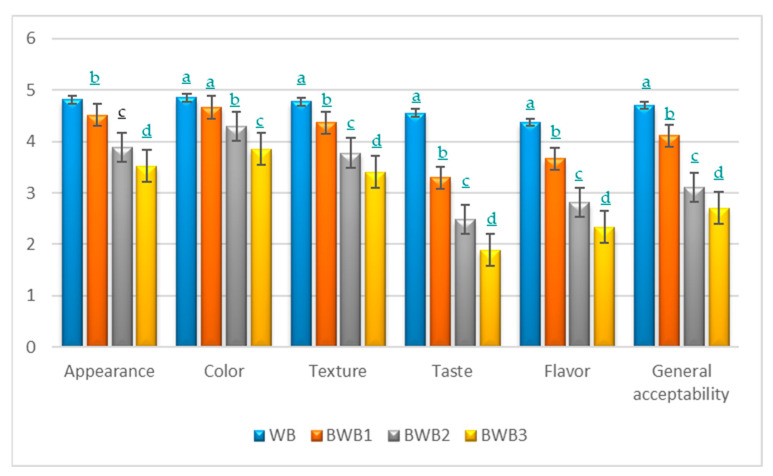
Global values of the sensory evaluation (consumer acceptance) of bread with baobab: WB (control bread); BWB1 (10% baobab flour and 90%wheat flour); BWB2 (20% baobab flour and 80% wheat flour); BWB3 (30% baobab flour and 70% wheat flour) using a five-point hedonic scale (*n* = 27). The values in the figure represent the mean of three determinations ± standard deviation (SD). The different letters (a–d) shown in the columns for each category of characteristics represent statistically significant differences (*p* < 0.05) recorded using a *t*-test.

**Table 1 foods-12-02697-t001:** Recipe for bread with composite baobab flour.

Samples	Ingredients						
	Baobab Flour (g)	Wheat Flour Type 650 (g)	Yeast (g)	Salt (g)	Honey (g)	Oil (mL)	Water (mL)
WB	-	1000	50	20	30	80	800
BWB1	100	900	50	20	30	80	800
BWB2	200	800	50	20	30	80	800
BWB3	300	700	50	20	30	80	800

**Table 2 foods-12-02697-t002:** Proximate composition of baobab composite flours and breads.

Samples	Nutritional Characteristics					
	Moisture	Mineral Content	Proteins	Lipids	Carbohydrates	Energy Values
	(%)	(%)	(%)	(%)	(g/100 g)	(kcal/100 g)
**Composite flours**						
WF	10.80 ± 0.04 ^d^	0.33 ± 0.03 ^c^	11.26 ± 0.02 ^a^	1.33 ± 0.03 ^c^	76.28 ± 0.04	362.13 ± 0.16
BF	13.79 ± 0.01 ^a^	4.00 ± 0.02 ^a^	4.31 ± 0.05 ^d^	1.56 ± 0.02 ^a^	76.34 ± 0.06	336.62 ± 0.16
BWF1	11.39 ± 0.24 ^c,d^	0.66 ± 0.02 ^c^	11.05 ± 0.06 ^a^	1.40 ± 0.05 ^b^	75.50 ± 0.13	358.80 ± 0.22
BWF2	11.59 ± 0.01 ^b,c^	1.10 ± 0.18 ^b^	10.16 ± 0.05 ^b^	1.42 ± 0.03 ^b^	75.73 ± 0.17	356.35 ± 0.69
BWF3	11.80 ± 0.03 ^b^	1.21 ± 0.04 ^b^	9.80 ± 0.02 ^c^	1.45 ± 0.01 ^b^	75.74 ± 0.07	355.21 ± 0.23
**Breads**						
BW	35.20 ± 0.02 ^a^	0.96 ± 0.02 ^d^	10.67 ± 0.01 ^a^	5.19 ± 0.03 ^c^	47.98 ± 0.02	281.31 ± 0.2
BWB1	34.60 ± 0.04 ^b^	1.69 ± 0.01 ^c^	8.90 ± 0.01 ^b^	5.20 ± 0.05 ^c^	49.60 ± 0.36	280.84 ± 9.5
BWB2	34.66 ± 0.05 ^b^	1.95 ± 0.03 ^b^	8.51 ± 0.01 ^c^	5.26 ± 0.02 ^b^	49.63 ± 0.03	279.84 ± 0.11
BWB3	34.73 ± 0.04 ^b^	2.17 ± 0.05 ^a^	8.02 ± 0.04 ^d^	5.30 ± 0.02 ^a^	49.79 ± 0.8	278.93 ± 0.41

Values of the table represent the mean of three determinations ± standard deviation (SD). Different letters (a–d) in the same column of each sample category represent statistically significant differences (*p* < 0.05) recorded using a *t*-test.

**Table 3 foods-12-02697-t003:** The macro- and micronutrient contents of the wheat flour, the composite flours, and the bread obtained.

Samples	Macro- and Microelements Contents (mg/kg)								
	Cu	Ni	Zn	Fe	Mn	Ca	Mg	K	Na
Composite flours									
BF	8.04 ± 0.05 ^a^	0.598 ± 0.002 ^a^	14.90 ± 0.01 ^a^	155.14 ± 2.95 ^a^	4.84 ± 0.05 ^d^	1570.67 ± 29.67 ^a^	1066.73 ± 9.97 ^a^	13,276.47 ± 174 ^a^	143.19 ± 5.22 ^a^
WF	2.4 ± 0.002 ^c^	nd	11.40 ± 0.01 ^b^	12.13 ± 0.03 ^e^	8.69 ± 0.02 ^a^	166.22 ± 0.99 ^d^	273.15 ± 1.28 ^e^	1188.68 ± 101 ^e^	60.38 ± 0.76 ^d^
BWF1	2.88 ± 0.03 ^c^	0.22 ± 0.026 ^c^	11.79 ± 0.01 ^b^	17.89 ± 0.20 ^d^	6.84 ± 0.05 ^b^	246.12 ± 2.62 ^c^	431.81 ± 7.5 ^d^	1589.59 ± 164 ^d^	83.86 ± 2.04 ^c^
BWF2	3.4 ± 0.17 ^b^	0.267 ± 0.001 ^c^	12.21 ± 0.04 ^c^	33.37 ± 0.44 ^c^	6.61 ± 0.29 ^b^	307.95 ± 2.05 ^b^	553.78 ± 3.41 ^c^	3640.07 ± 135 ^c^	87.66 ± 1.95 ^c^
BWF3	3.96 ± 0.02 ^b^	0.526 ± 0.037 ^b^	12.35 ± 0.09 ^c^	47.14 ± 0.4 ^b^	5.86 ± 0.15 ^c^	329.15 ± 6.40 ^b^	573.33 ± 0.75 ^b^	4155.47 ± 147 ^b^	96.57 ± 0.62 ^b^
Composite breads									
WB	1.96 ± 0.03 ^c^	0.152 ± 0.002 ^c^	12.20 ± 0.09 ^a^	11.15 ± 0.03 ^d^	7.14 ± 0.17 ^a^	327.49 ± 1.05 ^d^	264.47 ± 2.91 ^d^	984.53 ± 13.42 ^d^	40.44 ± 0.57 ^b^
BWB1	2.19 ± 0.02 ^b,c^	0.164 ± 0.002 ^c^	12.21 ± 0.04 ^a^	19.83 ± 0.11 ^c^	6.57 ± 0.25 ^b^	343.13 ± 8.61 ^c^	343.13 ± 5.26 ^c^	1392.13 ± 127 ^c^	45.39 ± 1.41 ^b^
BWB2	2.52 ± 0.03 ^a,b^	0.265 ± 0.002 ^b^	12.28 ± 0.02 ^a^	24.48 ± 0.36 ^b^	5.16 ± 0.15 ^c^	366.05 ± 3.55 ^b^	436.89 ± 5.34 ^b^	1686.30 ± 185 ^b^	55.79 ± 0.57 ^a^
BWB3	2.65 ± 0.02 ^a^	0.366 ± 0.007 ^a^	12.3 ± 0.05 ^a^	37.54 ± 0.64 ^a^	4.67 ± 0.06 ^d^	395.05 ± 2.85 ^a^	464.78 ± 1.94 ^a^	2059.74 ± 75 ^a^	56.57 ± 0.98 ^a^

The values of the table represent the mean of three determinations ± standard deviation (SD). Different letters (a–e) in the same column of each sample category represent statistically significant differences (*p* < 0.05), recorded using a *t*-test.

**Table 4 foods-12-02697-t004:** Bread quality indicators for control wheat bread (WB); BWB1 (10% baobab flour and 90% wheat flour); BWB2 (20% baobab flour and 80% wheat flour); and BWB3 (30% baobab flour and 70% wheat flour).

Indicator	MU	WB	BWB1	BWB2	BWB3
Volume	cm^3^/100 g	190.97 ± 1.05 ^a^	155.04 ± 0.95 ^b^	152.34 ± 1.02 ^b^	121.88 ± 1.16 ^c^
Porosity	%	78.84 ± 0.56 ^a^	73.50 ± 0.67 ^b^	70.53 ± 0.85 ^c^	66.22 ± 0.45 ^d^
Elasticity	%	90 ± 0.15 ^a^	85 ± 0.27 ^b^	75 ± 0.33 ^c^	72 ± 0.19 ^c^
H/D	-	0.62 ± 0.01 ^a^	0.58 ± 0.02 ^b^	0.57 ± 0.01 ^c^	0.55 ± 0.03 ^d^
Acidity	grade	2.4 ± 0.05 ^d^	5.4 ± 0.08 ^c^	10.8 ± 0.05 ^b^	15 ± 0.06 ^a^

Values of the table represent the mean of three determinations ± standard deviation (SD). Different letters (a–d) in the same row represent statistically significant differences (*p* < 0.05) recorded using a *t*-test.

**Table 5 foods-12-02697-t005:** Primary parameters of baobab composite flours.

Samples	WA (%)	ST (min)	C1	C2	C3	C4	C5	α (Nm/min)	β (Nm/min)	γ (Nm/min)
WF	55.8	9.52	1.189	0.538	2.027	1.860	3.289	−0.074	0.306	−0.046
BF	76.7	0.30	1.255	0.731	1.170	0.266	0.487	nd	0.158	−0.064
BWF 1	62.1	5.82	1.165	0.304	2.030	1.819	3.054	−0.094	0.568	−0.028
BWF 2	63.5	3.97	1.172	0.287	1.852	1.658	2.565	−0.090	0.532	−0.012
BWF 3	63.8	3.82	1.372	0.342	1.706	1.524	2.306	−0.110	0.530	−0.056

**Table 6 foods-12-02697-t006:** Mixolab Profiler index.

Samples	WAI	MI	GI	VI	AI	RI
WF	2	6	4	7	8	8
BWF 1	8	5	0	6	7	7
BWF 2	8	5	0	4	8	6
BWF 3	8	5	0	2	7	5

**Table 7 foods-12-02697-t007:** Phytochemical profile of baobab composite flours and bread.

Samples	Total Polyphenols Content (mg/100 g)	Total Flavonoids Content (mg/100 g)	Antioxidant Activity, DPPH (%)
**Flours**			
*WF*	176.7 ± 0.69 ^d^	1.72 ± 0.08 ^e^	31.77 ± 0.43 ^d^
*BF*	629.7 ± 0.35 ^a^	213.13 ± 0.08 ^a^	86.62 ± 0.04 ^a^
BWF1	157.97 ± 1.53 ^e^	70.53 ± 0.08 ^d^	82.08 ± 0.04 ^c^
BWF2	231.78 ± 1.97 ^c^	119.01 ± 0.1 ^c^	82.86 ± 0.56 ^c^
BWF3	296.1 ± 3.29 ^b^	138.97 ± 0.81 ^b^	84.86 ± 0.01 ^b^
**Breads**			
WB	183.8 ± 6.06 ^d^	2.90 ± 0.05 ^d^	31.003 ± 0.09 ^d^
BWB1	193.3 ± 2.1 ^c^	44.05 ± 0.81 ^c^	58.86 ± 0.01 ^c^
BWB2	232.1 ± 0.6 ^b^	111.68 ± 8.135 ^b^	61.66 ± 0.02 ^b^
BWB3	297.63 ± 1.75 ^a^	208.06 ± 0.002 ^a^	66.72 ± 0.07 ^a^

The values of the table represent the mean of three determinations ± standard deviation (SD). Different letters (a–e) in the same column of each sample category represent statistically significant differences (*p* < 0.05) recorded using a *t*-test.

## Data Availability

The data presented in this study are available on request from the corresponding author.

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
