# Peer review of "Nutritional, Physico-Chemical, Phytochemical, and Rheological Characteristics of Composite Flour Substituted by Baobab Pulp Flour (Adansonia digitata L.) for Bread Making"

_foods, 2023, doi:10.3390/foods12142697_

Round 1

Reviewer 1 Report

The manuscript has investigated the nutritional, physico-chemical, phytochemical, and rheological characteristics of composite flour substituted by baobab pulp flour for bread making. The topic is interesting; However, the manuscript has several problems:

1. L 50; "L." shouldn't be in italics.

2. Mention Materials as an individual part.

3. Mention Figure 1 in the text.

4. Part 2.5; Briefly, express each method.

5. L 119; Please add the reference.

6. Table 2; Change the title in English. "F" should be changed to "BF". Also, mention the significant letters for carbohydrates and energy values. In addition, "a" should be used for the highest value; check all tables.

7. L 202; Add the mineral content to the table. And how was it measured?

8. L 240; Number this part and mention the mineral content of the samples in this section. 

9. First, mention the pull name of macro and microelements, then their abbreviations. Check the entire manuscript. 

10. Figure 3; If it is possible provide a better picture of BWF2.

11. Figure 4; Mention the significant letters.

The manuscript should be edited by a professional English editor.

Author Response

Date: Timisoara 06.07.2023

Name: Negrea Monica

University:       University of Life Sciences „King Mihai I’’ from Timisoara, Calea Aradului No.119, 300645, Timisoara, Romania

Address: Calea Aradului No. 119, 300641 Timisoara, Romania

E-mail: monicanegrea@usvt.ro

Dear Reviewer,

We would like to address all our thanks and gratitude for the constructive observations, corrections and recommendations.

Based on the reviewers’ recommendations, the authors of this paper responded point by point to the following aspects:

  1. Comments: The manuscript has investigated the nutritional, physico-chemical, phytochemical, and rheological characteristics of composite flour substituted by baobab pulp flour for bread making. The topic is interesting; However, the manuscript has several problems:

 L 50; "L." shouldn't be in italics.

Answer: the correction was done

  1. Comments: Mention Materials as an individual part

Answer: the correction was done

  1. Comments: Mention Figure 1 in the text.

Answer: the correction was done in Line 88 we mentioned the figure 1 .

4.Comments: Part 2.5; Briefly, express each method.

Answer : we completed with a short description of each parameter determined

5.Comments: L 119; Please add the reference

Answer: the references was added

  1. Comments: Table 2; Change the title in English. "F" should be changed to "BF". Also, mention the significant letters for carbohydrates and energy values.- In addition, "a" should be used for the highest value; check all tables.

Answer: For carbohydrates and energy value no statistically significant difference can be made because these values are calculated according to the averages of the other values (protein, fat, moisture, ash). The other correction was done

  1. Comments: L 202; Add the mineral content to the table. And how was it measured?

Answer: we changed ash with mineral content, the results are reported as %

  1. Comments: L 240; Number this part and mention the mineral content of the samples in this section. 

Answer: we numbered this section 3.2.

  1. Comments: First, mention the pull name of macro and microelements, then their abbreviations. Check the entire manuscript. 

Answer: we mentioned the pull name of macro and microelements and then their abbreviation in the entire manuscript

10.Comments: Figure 3; If it is possible provide a better picture of BWF2.-

 Answer: the correction was done

  1. 11. Comments: Figure 4; Mention the significant letters.

Answer:the correction was done

Once again, we would like to thank the reviewer for your appreciations, corrections and recommendations which contributed to the significant improvement of the paper.

Reviewer 2 Report

Introduction

1.  Line 41-43: rewrite for clarity, does not look a good sentence structure. Not everything is bioactive protein for instance, phenolic compounds and antioxidants are two different things.

2. Lines 53, 55, 60-61, 62: You mention Baobab to be rich in vitamins and minerals several times. Please refrain from repeating same information.

3. Can you add few researches where products are made using baobab plant. There are few available in the literature.

4. Lines 71-72: again repeat of same information as in line 69-70.

Materials and Methods

1. Lines 81-82: the bread was prepared instead of will be prepared. Lines 92-93 use past tense and not future tense.

2. Line 88: the pre-culturing process is possibly not called fermentation. Please check

3. Line 90: What is first and second speed? Mention rpm instead of first and second speed.

4. how were the sample prepared (extraction process) for analysis of TPC and Antioxidant activity?

5. Rheological characterization method described is very vague. Please provide more details so that the results can be reproduced by any other group conducting experiments following this protocol.

6. Line 168-139- delete repeated sentence.

Results and Discussions:

1. Figures can be improved, remove gridlines, ensure uniformity in size and dimension. The axis values should be sufficiently large for readers to understand easily. Figure 3 and 4.

2. In most discussions, author's own verdict is missing. The study is compared to existing literature however no critical reason for this specific finding is put forward. For example, in sensory study, the samples with higher baobab constituent are found to be have lower sensory score however the authors own insight to why this is happening is missing. They could mention what is causing this lower acceptability by sensory panelists? does baobab have a characteristics taste/flavour on its own and its addition in higher quantities affect overall taste/flavour? what is the reason for this low score?

Similar approach should be used while discussing other results too. First put the results/finding in this study, followed by comparison to existing research then it should follow author's own insight to why this phenomenon is unique and what is the reason behind this finding.

Check references for uniformity and keep them as journal standard. Some have links while others do not have doi links.

English must be improved throughout the manuscript for clarity and ease of comprehension. There are several grammatical errors and sentences lack cohesiveness that make understanding very difficult. For example:

1.Lines 81-82, 92-93, 95: future tense is used instead of past to start the methodology section.

2.. Line 113-determination of the .....was determined by

3. Lines 154-155: this sentence is not clear. Please rewrite to make it clear and meaningful.

While sentence structuring is done quite well in most cases, there are also places where English clarity is difficult to make any sense. I suggest a full proofread of the article and correct English for clarity and comprehension.

Author Response

Date: Timisoara 06.07.2023

Name: Negrea Monica

University:       University of Life Sciences „King Mihai I’’ from Timisoara, Calea Aradului No.119, 300645, Timisoara, Romania

Address: Calea Aradului No. 119, 300641 Timisoara, Romania

E-mail: monicanegrea@usvt.ro

Dear Reviewer,

We would like to address all our thanks and gratitude for the constructive observations, corrections and recommendations.

Based on the reviewers’ recommendations, the authors of this paper responded point by point to the following aspects:

Introduction

  1. Comments: Line 41-43: rewrite for clarity, does not look a good sentence structure. Not everything is bioactive protein for instance, phenolic compounds and antioxidants are two different things.

        Answer:– we reformulated the sentence

  1. Comments: Lines 53, 55, 60-61, 62: You mention Baobab to be rich in vitamins and minerals several times. Please refrain from repeating same information.

Answer: we reformulated the sentences so that the information is not repeated

  1. Comments: Can you add few researches where products are made using baobab plant. There are few available in the literature

Answer: we added references as you suggested 

  1. Comments: Lines 71-72: again repeat of same information as in line 69-70.

Answer: we deleted the repeated sentence in line 71-72.

Materials and Methods

  1. Comments: Lines 81-82: the bread was prepared instead of will be prepared. Lines 92-93 use past tense and not future tense.

Answer: we changed and used past tense

  1. Comments: Line 88: the pre-culturing process is possibly not called fermentation. Please check –

Answer: the correction was done

  1. Comments: Line 90: What is first and second speed? Mention rpm instead of first and second speed.

Answer: the correction was done

  1. Comments: how were the sample prepared (extraction process) for analysis of TPC and Antioxidant activity?

Answer: the description of the sample preparation was inserted in the manuscript

  1. Comments: Rheological characterization method described is very vague. Please provide more details so that the results can be reproduced by any other group conducting experiments following this protocol.

Answer: we completed the description of the rheological determination.

  1. Comments: Line 168-139- delete repeated sentence.

Answer: we deleted the sentence

Results and Discussions:

1.Comments: Figures can be improved, remove gridlines, ensure uniformity in size and dimension. The axis values should be sufficiently large for readers to understand easily. Figure 3 and 4.

Answer: we improved the figures as requested

  1. Comments: In most discussions, author's own verdict is missing. The study is compared to existing literature however no critical reason for this specific finding is put forward. For example, in sensory study, the samples with higher baobab constituent are found to be have lower sensory score however the authors own insight to why this is happening is missing. They could mention what is causing this lower acceptability by sensory panelists? does baobab have a characteristics taste/flavour on its own and its addition in higher quantities affect overall taste/flavour? what is the reason for this low score?

Answer: we completed the sensory study with authors own verdict

Comments: Similar approach should be used while discussing other results too. First put the results/finding in this study, followed by comparison to existing research then it should follow author's own insight to why this phenomenon is unique and what is the reason behind this finding.

Answer: we completed the results with authors own verdict

Comments: Check references for uniformity and keep them as journal standard. Some have links while others do not have doi links.

Answer: The references were standardized

Comments on the Quality of English Language

English must be improved throughout the manuscript for clarity and ease of comprehension. There are several grammatical errors and sentences lack cohesiveness that make understanding very difficult. For example:

1.Comments: Lines 81-82, 92-93, 95: future tense is used instead of past to start the methodology section.- Answer: we changed the future tense and used past tense

2.Comments: Line 113-determination of the .....was determined by

Answer: the sentence was reformulated

3.Comments: Lines 154-155: this sentence is not clear. Please rewrite to make it clear and meaningful.

Answer: we reformulated the sentence so that it has a clearer meaning

Comments: While sentence structuring is done quite well in most cases, there are also places where English clarity is difficult to make any sense. I suggest a full proofread of the article and correct English for clarity and comprehension.

Answer: We have read the manuscript corrected the sentences in terms of English

Once again, we would like to thank the reviewer for your appreciations, corrections and recommendations which contributed to the significant improvement of the paper.
